# WAVELET-GUIDED ATTENTION NETWORK FOR SPIKING IMAGE RECONSTRUCTION

## ABSTRACT

Reconstructing high-quality images from spike data remains a challenging task, particularly under low-light and high-motion conditions where spike noise and motion blur are prominent. To tackle these challenges, we propose WaveAttNet, a novel Wavelet-Guided Attention Network for spiking image reconstruction. WaveAttNet comprises two ky components: (1) a Wavelet-Guided Attention (WGA) module that performs frequency-aware noise suppression by emphasizing informative subbands and suppressing noisy ones through discrete wavelet transform and attention weighting; and (2) a Multi-scale Temporal Attention (MTA) module that captures and fuses temporal features across multiple time scales to mitigate both short-exposure noise and long-exposure motion blur. Extensive experiments on both synthetic (Spike-REDS) and real-world (Real-Captured) spike datasets demonstrate that WaveAttNet outperforms state-of-the-art approaches in terms of perceptual quality and quantitative metrics.

## 1 INTRODUCTION

Spike cameras (Huang et al. (2023)), inspired by the asynchronous processing of biological vision systems, offer several distinct advantages over traditional frame-based sensors. These include ultra-high temporal resolution, high dynamic range, and low power consumption (Gehrig et al. (2018); Maqueda et al. (2018)). Such properties enable spike cameras to capture high-speed motion and operate reliably under dynamic lighting conditions, scenarios that are often challenging for conventional vision systems.

Extensive research has been devoted to spike image reconstruction, which can be broadly categorized into early model-based approaches and learning-based methods. Early model-based methods, such as TFP (Zhu et al. (2019)) and TFI (Zhu et al. (2019)), rely on the physical principles of spike generation and focus on interpreting spike timing statistics. While these methods are interpretable and computationally lightweight, they are often vulnerable to motion blur and noise amplification, particularly under low-light conditions where spike activity is sparse and irregular. To enhance spike image reconstruction, researchers have increasingly turned to learning-based methods, especially those based on convolutional neural networks (CNNs) and Transformers. CNN-based models (e.g., Spk2ImgNet (Zhao et al. (2021)), WGSE (Zhang et al. (2023))) leverage hierarchical spatial-temporal feature extraction to enhance reconstruction quality and robustness. In contrast, Transformer-based models (e.g., SpikeFormer (She & Qing (2022)), SwinSF (Jiang et al. (2024))) utilize attention mechanisms to capture long-range dependencies and fine-grained temporal dynamics. Both approaches have demonstrated substantial performance gains over traditional approaches.

However, reconstructing high-quality images from spike streams remains challenging due to inherent limitations in spike sensing and the accumulations process: (1) Noise under short exposures (Zhang et al. (2023)). In low-light conditions, short exposure periods lead to low photon counts and high variability in light arrival, resulting in significant noise. Most reconstruction methods struggle to handle such noise effectively, resulting in images with considerable noise and reduced brightness. (2) Motion blur under long exposures (Yin et al. (2025)). To suppress noise, longer exposure periods are often used. However, in scenes with fast-moving objects, this leads to motion blur. As objects change position over time, their spikes are spatially accumulated, resulting in blurred edges, ghosting artifacts, and distorted shapes in the reconstructed image.

To overcome these challenges, we propose WavelAttNet, a novel wavelet attention-based network for spike image reconstruction. It integrates two key components: First, a wavelet-guided attention module that effectively suppresses noise and enhances temporal feature representation. This module applies DWT to decompose the spike stream into multi-scale wavelet coefficients, naturally separating the signal into different frequency bands. This separation helps distinguish informative high-frequency features from high-frequency noise. The attention mechanism further strengthens meaningful coefficients while filtering out irrelevant or noisy ones. Second, an attention-based multi-scale temporal fusion module that handles motion blur and low-light issues by aggregating spike features across multiple temporal scales. Short exposure windows preserve fine details and fast motion but often suffer from noise and weak signals in low-light conditions. In contrast, long exposure windows reduce noise through temporal averaging but introduces motion blur. To balance these trade-offs, the attention mechanism enables the model to adaptively focus on the most informative features–emphasizing short-scale features in dynamic regions and long-scale features in noisy, low-light area–resulting in improved overall image quality. The main contribution of this work can be summarized as follows:

- We propose a novel wavelet attention-based network for spike image reconstruction, where a wavelet-guided attention (WGA) module is introduced to effectively suppress noise and enhance temporal feature representation.
- We introduce a multi-scale temporal attention (MTA) module that addresses motion blur and low-light challenges through attention mechanisms that capture informative dynamics across varying temporal scales.
- Extensive experiments on the Spike-REDS and Real-Captured benchmarks consistently demonstrate that our method outperforms state-of-the-art baselines across multiple evaluation metrics, delivering higher reconstruction accuracy, sharper structural details, and superior perceptual quality.

## 2 RELATED WORK

### 2.1 SPIKE CAMERAS IN VISUAL TASKS

Spike cameras, as bio-inspired neuromorphic sensors, have enabled breakthroughs in various visual tasks (Yang et al. (2024); Dong et al. (2024); Chang et al. (2024)) by leveraging their ultra-high temporal resolution, low latency, and wide dynamic range. Unlike conventional frame-based cameras that capture images at fixed intervals, spike cameras operate asynchronously, recording absolute light intensity at each pixel by firing binary spikes whenever the accumulated light exceeds a predefined threshold. This continuous, event driven sensing allows for high-frequency, low-power operation, making spike cameras particularly attractive for capturing dynamic scenes with high-speed motion or changing illumination.

In recent years, spike cameras have been increasingly applied to low-level vision tasks, especially image reconstruction. However, despite substantial progress, spike data presents several challenges that complicate the reconstruction process. One major issue is noise, especially in low-light conditions (Chen et al. (2025); Zhu et al. (2023)). When the lighting is limited, the sensor captures only a small number of photons, leading to sparse and unreliable spike outputs. This results in reconstructed images that appear noisy, with low contrast and poor texture details. Another challenge is motion blur in high-speed scenes (Yin et al. (2025); Zhang et al. (2023)). To reduce noise, longer exposure windows are often used, but this causes problems when objects move quickly. As a result, motion gets averaged across multiple pixels, leading to blurry edges and ghosting effects in the reconstructed images. These challenges underscore the need for adaptive, context-aware reconstruction methods that can effectively handle the noise, motion blur, and complex spatiotemporal patterns inherent in spike data, which result from its asynchronous and random nature.

### 2.2 SPIKE-TO-IMAGE RECONSTRUCTION

Spike image reconstruction, a fundamental task for neuromorphic spike cameras, aims to recover high-quality intensity images from dense, asynchronous binary spike streams. Over the years, research in this area has evolved from model-driven physical methods to data-driven deep learning

approaches, with efforts to address key challenges such as high-speed motion blur, noise, and low-light conditions.

Early model-driven methods–such as TFP (Zhu et al. (2019)) and TFI (Zhu et al. (2019))–rely on the underlying physical principles of spike generation to estimate light intensity, offering strong interpretability and no need for large labeled datasets. However, they struggle to handle complex noise (especially in low-light environments) and fast, irregular motion, often resulting in blurry or unstable reconstructions. In contrast, learning-based methods utilize deep neural networks to learn the mapping from spike streams to intensity images, enabling more effective feature extraction from data. These methods can be broadly categorized by their backbone architectural backbones, including CNN-based networks (e.g. Spk2ImgNet (Zhao et al. (2021)), WGSE (Zhang et al. (2023)), RSIR (Zhu et al. (2023))), transformer-based models (e.g., SwinSF (Jiang et al. (2024)), Spikeformer (She & Qing (2022))), and Mamba-based architectures (e.g. Spk2ImgMamba (Yin et al. (2025))). Through ongoing innovations in network design, learning-based approaches continue to improve in terms of accuracy and adaptability, while still facing trade-offs between precision, efficiency, and robustness to real-world variability.

## 3 METHODOLOGY

Fig. 1 illustrates the overview of WaveAttNet, a novel wavelet-guided attention network designed for spike image reconstruction. The architecture comprises two key components: First, the wavelet-guided attention module is introduced to effectively suppress temporal noise and enhance temporal feature representation. Second, the multi-scale temporal attention module tackles motion blur and low-light challenges by selectively capturing informative features across multiple temporal scales, enhancing the model's generalization to real-world scenarios.

### 3.1 WAVELET-GUIDED ATTENTION (WGA) MODULE

The motivation behind the wavelet-guided attention module is to first apply multi-level DWT to decompose the input signal into multi-level low- and high-frequency components, allowing informative high-frequency features (such as edges and textures) to be separated from high-frequency noise. An attention mechanism is then used to selectively enhance meaningful subbands while suppressing irrelevant or noisy ones.

Let the input spike stream be $S \in \mathbb{R}^{T \times W \times H}$, where $T$ is the number of time steps, and $W, H$ denote spatial dimensions. At each spatial location $(i, j)$, the spike sequence over time is defined as:

$$S(i, j) = \{S_t(i, j)\}_{t=1}^{T} \in \mathbb{R}^{T}$$

We apply a multi-level DWT along the temporal axis at each pixel location to decompose the signal into multi-level low- and high-frequency components.

**First-Level Decomposition.** At pixel $(i, j)$, the 1D spike signal $S_{i,j}$ is filtered with low-pass $D_L$ and high-pass $D_H$ filters:

$$S_{L1} = (2 \downarrow)(D_L * S_{ij}), \quad S_{H1} = (2 \downarrow)(D_H * S_{ij})$$

where $*$ denotes convolution and $(2 \downarrow)$ indicating downsampling by a factor of 2.

**Multi-Level Decomposition.** Low-frequency outputs from each level are recursively decomposed:

$$S_{Lk} = (2 \downarrow)(D_L * S_{L(k-1)}), \quad S_{Hk} = (2 \downarrow)(D_H * S_{L(k-1)})$$

where $k$ indicates the $k$-th level decomposition. After $N$ levels, the full coefficient/subband set is:

$$\mathcal{W}(S) = \{S_{H1}, S_{H2}, ..., S_{HN}, S_{LN}\}$$

Here, $S_{Hn}(n = 1, ..., N)$ are the high-frequency coefficients, and $S_{LN}$ is the low-frequency coefficient from the last decomposition level.

To adaptively weigh the importance of each subband, we use an attention mechanism that highlights useful subbands and suppresses less relevant ones. Denote the subbands as:

$$\mathcal{W}(S) = \{S_l \in \mathbb{R}^{T_l}\}_{l=1}^{N+1}$$

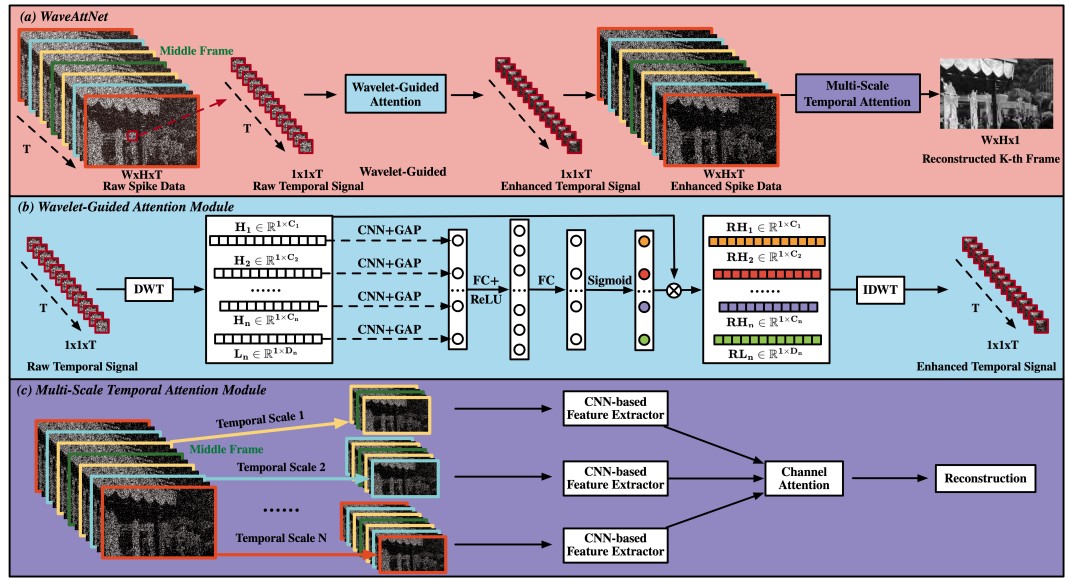

Figure 1: Overview of (a) WaveAttNet, which includes a (b) wavelet-guided attention module for noise reduction and temporal-frequency feature enhancement, and a (c) multi-scale temporal attention module designed to capture informative dynamics across varying temporal scales, effectively addressing motion blur and low-light challenges.

where $l = 1, ..., N$ are high-frequency subbands, and $l = N + 1$ is the low-frequency subband. We extract features from each subband using a shallow CNN, then compute a global descriptor for each subband through average pooling:

$$g_l = \frac{1}{T_l} \sum_{t=1}^{T_l} S_l(t)$$

These descriptors are passed through two fully connected layers with a ReLU activation, followed by a sigmoid function to produce the attention weights:

$$\mathbf{w} = Softmax(FC(ReLU(FC(\mathbf{g})))) \tag{1}$$

where $\mathbf{w} = [w_1, w_2, \ldots, w_{N+1}] \in \mathbb{R}^{N+1}$ represents the attention weights for each subband.

Next, each wavelet subband is scaled by its corresponding weight:

$$\hat{S}_l = w_l \cdot S_l \quad l = 1, \ldots, N + 1$$

The weighted subbands are reconstructed into spike streams using inverse DWT, resulting noise-reduced and enhanced feature representations. This attention mechanism enables the model to focus on important frequency components while effectively suppressing noise.

## 3.2 MULTI-SCALE TEMPORAL ATTENTION (MTA) MODULE

Multi-scale temporal feature extraction offers significant advantages over single-scale approaches for spike data by capturing a broader range of temporal dynamics, which is critical for real-world scenarios. We propose a multi-scale temporal attention module to tackle two major challenges in spike-based imaging: First, noise under short exposures. In low-light conditions, short exposure durations lead to fewer photons and high variability, resulting in noisy spike data. Second, motion blur under long exposures. Although longer exposures reduce noise, they often blur fast-moving objects because spike events from different positions accumulate over time, leading to smeared edges and ghosting artifacts. To handle both problems, the MTA module use multiple parallel branches, each processing a short-term spike segment centered on the same temporal midpoint but with a

different temporal window size. This design allows the model to balance noise reduction and motion preservation by learning from multiple temporal perspectives.

Let the input spike stream be $S = \{S_t \in \mathbb{R}^{W \times H}\}_{t=1}^{T}$, where $T$ is the number of time steps, and define the central temporal index as $t_c = \lfloor \frac{T}{2} \rfloor$. The module consists of $N$ parallel branches, each extracting features from a distinct short-term temporal window centered at $t_c$, but with varying temporal lengths. It operates in several sequential stages:

**Temporal Segments at Different Scales.** For each branch $n \in \{1, 2, \ldots, N\}$, we define a temporal window length $T_n$. The spike segment for branch $n$ is:

$$S^{(n)} = \left\{ S_{t_c - \frac{T_n}{2}}, \ldots, S_{t_c + \frac{T_n}{2}} \right\} \in \mathbb{R}^{T_n \times W \times H}$$

Each branch processes its corresponding temporal segment $S^{(n)}$, allowing the network to capture distinct and complementary temporal features across different time scales.

**Feature Extraction Per Branch.** Each branch includes an independent CNN-based extractor $f^{(n)}$ that extracts temporal-spatial features:

$$F^{(n)} = f^{(n)}(S^{(n)})$$

**Attention Weighting Across Scales.** To adaptively weigh the contribution of each temporal scale, an attention map $\alpha_n$ is computed for each branch. This is achieved using two convolutional layers with ReLU and softmax activations, respectively:

$$\alpha_n = Softmax(Conv_2(ReLU(Conv_1(F^{(n)}))))$$

The attention map $\alpha_n$ modulates the feature response in each branch. The final multi-scale temporal feature $F_{\text{out}}$ is obtained by applying the learned attention weight to each branch independently:

$$F_{\text{out}} = \{\alpha_n \cdot F^{(n)}\} \quad n = 1, \ldots, N$$

The outputs from all branches are then be concatenated and passed to the downstream reconstruction module.

## 4 EXPERIMENTAL SETUP

In this section, we present the experimental setup, covering the datasets, evaluation metrics, and implementation details.

### 4.1 DATASETS

For training, we use the widely adopted synthesized Spike-REDS dataset (Zhao et al. (2021)), which provides both spike streams and corresponding ground truth images at a resolution of 250×400. To evaluate the generalization ability of WaveAttNet, we conduct testing on both the test set of Spike-REDS and the diverse Real-Captured spike dataset previously used in (Yin et al. (2025)). The Real-Captured dataset includes sequences from the recVidarReal2019 (Zhu et al. (2020)) and momVidarReal2021 (Zheng et al. (2023)) datasets, featuring a resolution of 400×250. This dataset contains challenging high-speed motion scenarios under complex indoor and outdoor conditions, providing a robust benchmark for assessing generalization performance.

### 4.2 EVALUATION METRICS.

On the paired Spike-REDS dataset, which provides both spike data and corresponding ground truth images, we evaluate performance using three commonly used full-reference metrics: PSNR, SSIM, and LPIPS (Zhang et al. (2018)). For the Real-Captured dataset, where ground truth images are unavailable, we rely on no-reference metrics–NIQE (Mittal et al. (2012b)) and BRISQUE (Mittal et al. (2012a))–for evaluation. Additionally, we provide a more comprehensive comparison by reporting model size (in millions of parameters) and computational cost (in GFLOPs).

### 4.3 IMPLEMENTATION DETAILS

The WaveAttNet framework is implemented in PyTorch using a convolutional neural network (CNN) architecture. Training is performed on a workstation with an NVIDIA RTX 4090 GPU and an Intel Xeon Platinum 8470Q CPU. A batch size of 16 is used. The model is optimized using Adam, with an initial learning rate of 1e-4, which is reduced to 1e-5 after 25 epochs.

Table 1: Quantitative performance comparison of various spike image reconstruction methods on Spike-REDS.

| Methods | Backbones | Params (M) | FLOPs (G) | PSNR↑ | SSIM↑ | LPIPS↓ |
|---|---|---|---|---|---|---|
| **TFP** | – | – | – | 22.37 | 0.5801 | 0.3035 |
| **TFI** | – | – | – | 24.94 | 0.7150 | 0.3716 |
| **TFSTP** | – | – | – | 22.37 | 0.7300 | 0.3254 |
| **WGSE** | CNN | 3.81 | 415.26 | 38.88 | 0.9774 | 0.0212 |
| **Spk2ImgNet** | CNN | 3.76 | 1000.5 | 38.44 | 0.9767 | 0.0229 |
| **SSIR** | SNN | 0.382 | 23.8 | 16.32 | 0.8451 | 0.1234 |
| **SwinSF** | Transformer | 1.8 | 415.82 | 39.34 | 0.9803 | 0.0184 |
| **SpikeFormer** | Transformer | 7.58 | 67.88 | 37.18 | 0.9738 | 0.0580 |
| **Spk2ImgMamba** | Mamba | 0.282 | 9.9 | 15.15 | 0.8463 | 0.1248 |
| **WaveAttNet** | CNN | 3.78 | 1037.73 | **39.35** | **0.9810** | **0.0178** |

Table 2: Quantitative performance comparison of various spike image reconstruction methods on Real-Captured.

| Methods | Backbones | Params (M) | FLOPs (G) | NIQE↓ | BRISQUE↓ |
|---|---|---|---|---|---|
| **TFP** | – | – | – | 21.7393 | 82.0609 |
| **TFI** | – | – | – | 16.1018 | 63.9647 |
| **TFSTP** | – | – | – | 52.3297 | 83.1865 |
| **WGSE** | CNN | 3.81 | 415.26 | 8.3484 | 30.3575 |
| **Spk2ImgNet** | CNN | 3.76 | 1000.5 | 6.9135 | 32.7683 |
| **SSIR** | SNN | 0.382 | 23.8 | 7.4298 | 19.5845 |
| **SwinSF** | Transformer | 1.8 | 415.82 | 8.3614 | 26.7702 |
| **SpikeFormer** | Transformer | 7.58 | 67.88 | 6.3051 | 30.4309 |
| **Spk2ImgMamba** | Mamba | 0.282 | 9.9 | 6.7393 | **17.5177** |
| **WaveAttNet** | CNN | 3.78 | 1037.73 | **6.3031** | 21.6980 |

## 5 RESULTS AND DISCUSSIONS

### 5.1 COMPARISON WITH STATE-OF-THE-ART METHODS

We conduct an extensive comparison of the proposed WaveAttNet against a wide range of state-of-the-art approaches, including traditional model-based methods (e.g., TFP (Zhu et al. (2019)), TFI (Zhu et al. (2019)), TFSTP (Zheng et al. (2021))), CNN-based frameworks (e.g., Spk2ImgNet (Zhao et al. (2021)), WGSE (Zhang et al. (2023))), SNN-based models (e.g., SSIR (Zhao et al. (2023))), Transformer-based approaches (e.g., SpikeFormer (She & Qing (2022)), SwinSF (Jiang et al. (2024))), and the recent Mamba-based model (e.g., Spk2ImgMamba (Yin et al. (2025))).

Tables 1 and 2 present the quantitative comparison of various spike image reconstruction methods on the Spike-REDS and Real-Captured datasets, repectively. The proposed WaveAttNet achieves the best performance across all three full-reference metrics (PSNR, SSIM, and LPIPS) on Spike-REDS, and also records the best NIQE score on the Real-Captured dataset. These results highlight the effectiveness of our wavelet-guided attention module in suppressing noise through frequency-

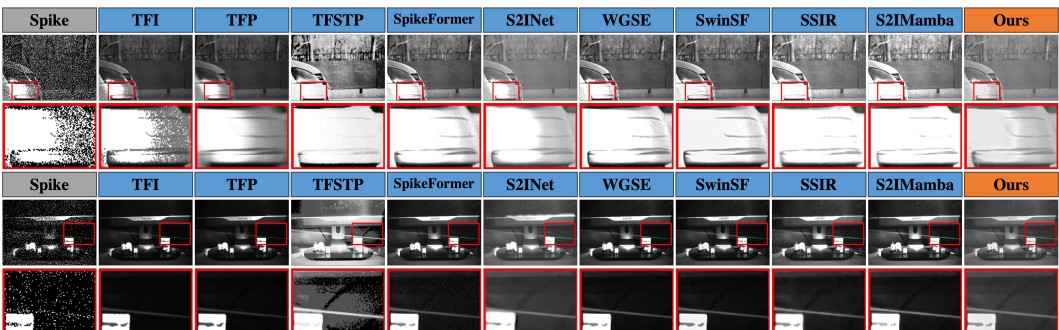

Figure 2: Visual comparison of spike image reconstruction methods on the Spike-REDS dataset.

Figure 3: Visual comparison of spike image reconstruction methods on the Real-Captured dataset.

aware filtering and the multi-scale temporal attention module in capturing temporal dynamics, which together improve generalization to low-light and fast-motion scenes.

However, WaveAttNet lags behind Spk2ImgMamba and SSIR in terms of the BRISQUE score. We believe this is due to BRISQUE's bias toward natural image statistics. As a no-reference quality metric, BRISQUE favors images with typical luminance, texture, and statistical distributions commonly found in natural scenes. Spike-based reconstructed images often lack these natural details, especially after strong noise removal or exposure correction. Although WaveAttNet improves overall image quality and performs well on PSNR and SSIM, its filtering process may smooth out textures. This makes the images look less "natural" to BRISQUE, leading to lower scores.

In contrast, the SNN-based SSIR and Mamba-based Spk2ImgMamba models perform unexpectedly poorly on the Spike-REDS dataset, primary due to their tendency to generate overly bright images with unrealistic illumination that deviates significantly from the ground truth, as illustrated in Fig. 2. This performance gap is largely attributed to the lack of dedicated illumination handling mechanisms, such as the multi-scale temporal attention module, which adaptively adjusts the importance of temporal scales based on varying lighting conditions. As illustrated in Fig. 3, WaveAttNet produces more natural lighting and preserves finer image details.

Table 3: Ablation results of the wavelet-guided attention module on the Spike-REDS (REDS) and Real-Captured (REAL) datasets. 'CNN' and 'Attention' denote the CNN and the attention blocks within the module, respectively.

| Datasets | CNN | Attention | PSNR | SSIM | LPIPS | Datasets | CNN | Attention | NIQE | BRISQUE |
|---|---|---|---|---|---|---|---|---|---|---|
| **REDS** | ✗ | ✗ | 38.36 | 0.9707 | 0.0231 | **REAL** | ✗ | ✗ | 6.9006 | 32.9598 |
| | ✓ | ✗ | 38.96 | 0.9795 | 0.0193 | | ✓ | ✗ | 6.5456 | 25.3307 |
| | ✓ | ✓ | 39.35 | 0.9810 | 0.0178 | | ✓ | ✓ | 6.3031 | 21.6980 |

Table 4: Impact of wavelet decomposition levels (Decomp Level) on the performance of the Spike-REDS (REDS) and Real-Captured (REAL) datasets.

| Datasets | Decomp Level | PSNR | SSIM | LPIPS | Datasets | Decomp Level | NIQE | BRISQUE |
|---|---|---|---|---|---|---|---|---|
| **REDS** | 2 | 39.22 | 0.9791 | 0.0183 | **REAL** | 2 | 6.5450 | 23.0038 |
| | 3 | 39.31 | 0.9788 | 0.0181 | | 3 | 6.4087 | 22.0334 |
| | 4 | **39.35** | 0.9804 | 0.0179 | | 4 | 6.3192 | **21.5390** |
| | 5 | 39.35 | **0.9810** | **0.0178** | | 5 | **6.3031** | 21.6980 |

Table 5: Ablation results for the multi-scale temporal attention (MTA) module on the Spike-REDS (REDS) and Real-Captured (REAL) datasets.

| Datasets | MTA | PSNR | SSIM | LPIPS | Datasets | MTA | NIQE | BRISQUE |
|---|---|---|---|---|---|---|---|---|
| **REDS** | *w/o* MTA | 38.77 | 0.9780 | 0.0209 | **REAL** | *w/o* MTA | 6.6683 | 26.8290 |
| | *w/* MTA | 39.35 | 0.9810 | 0.0178 | | *w/* MTA | 6.3031 | 21.6980 |

## 5.2 ABLATION STUDY

### 5.2.1 EVALUATION OF THE WAVELET-GUIDED ATTENTION (WGA) MODULE.

The wavelet-guided attention (WGA) module consists of three key components: a Discrete Wavelet Transform (DWT) for frequency decomposition, an attention mechanism for noise suppression and frequency feature enhancement, and an Inverse DWT (IDWT) for spike stream reconstruction. The attention mechanism itself includes a CNN block for deep frequency feature extraction and an attention block for adaptive frequency weighting.

To assess the effectiveness of the WGA module, we conduct ablation studies using three model variants: (1) the full model with the complete WGA module, (2) the model without the WGA module, and (3) the model with WGA but excluding the attention block. The ablation results of the WGA module are presented in Table 3, from which we observe that incorporating CNN block improves performance, likely because it transforms raw wavelet coefficients into more expressive and informative representations. The attention block further boosts performance, as it adaptively reweighs the frequency subbands, highlighting informative frequency components such as motion and texture while suppressing noisy or irrelevant ones. In contrast, CNN block alone treats all features uniformly without such selective enhancement.

### 5.2.2 EVALUATION OF DECOMPOSITION LEVEL IN WGA.

The number of decomposition levels in the Wavelet-Guided Attention (WGA) module has a strong impact on reconstruction performance. With too few levels (e.g., 2), the model struggles to separate high-frequency noise from informative high-frequency features such as motion and texture. As shown in Table 4, performance improves as the level increases from 2 to 4 and then stabilizes. Excessive decomposition, however, cause oversmoothing and loss of fine motion details, especially in sparse spike streams. Therefore, we set the decomposition level to 5 in our experiments.

### 5.2.3 EVALUATION OF THE MULTI-SCALE TEMPORAL ATTENTION (MTA) MODULE

Table 5 presents the ablation results of WaveAttNet with ($w/$) and without ($w/o$) the proposed multi-scale temporal attention (MTA) module on the Spike-REDS and Real-Captured datasets. The model equipped with MTA consistently outperforms its counterpart without MTA, demonstrating the effectiveness of multi-scale temporal feature extraction for spike data. The improvements are attributed to MTA's ability to capture a wider range of temporal dynamics, which is essential in real-world conditions. The MTA module employs multiple parallel branches, each operating at a different temporal scale. The integrated attention mechanism further weighs the contributions from each temporal scale, enabling the model to effectively balance noise suppression and motion preservation by learning from diverse temporal contexts.

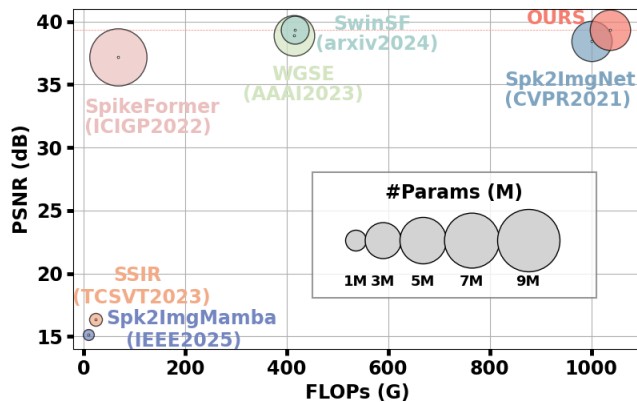

Figure 4: Quantitative performance (PSNR) and computational efficiency (FLOPs and model size) across representative spike image reconstruction models.

## 6 CONCLUSION

In this paper, we proposed WaveAttNet, a wavelet-guided attention network for spike image reconstruction that effectively addresses the challenges of temporal noise and motion blur. The proposed Wavelet-Guided Attention (WGA) module enhances informative frequency components while suppressing noise, and the Multi-scale Temporal Attention (MTA) module adaptively captures temporal dynamics at multiple scales to balance motion preservation and noise reduction. Extensive experiments demonstrate that WaveAttNet achieves state-of-the-art reconstruction performance across both synthetic and real-world spike datasets, outperforming existing approaches on PSNR, SSIM, LPIPS, and NIQE metrics.

However, as shown in Fig. 4, WaveAttNet has the largest model size and FLOPs among all compared methods. While this contributes to its strong performance, it also raises concerns about computational efficiency and deployment practicality. Therefore, improving model compactness and reducing computational overhead will be important directions for future work.

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

## A APPENDIX

### A.1 USE OF LLMS

Large Language Models (LLMs) were used solely to assist with writing and polishing the text.

## A.2 CODE OF ETHICS AND ETHICS STATEMENT

The research conducted in the paper conform, in every respect, with the ICLR Code of Ethics
`https://iclr.cc/public/CodeOfEthics`.

## A.3 REPRODUCIBILITY

This paper provides all necessary details to enable reproduction of the main experimental results, including dataset descriptions, training procedures, training parameters, and evaluation protocols.

