# OpenReview forum: "Wavelet-Guided Attention Network for Spiking Image Reconstruction"
_ICLR.cc/2026/Conference — ICLR 2026 Conference Withdrawn Submission_

### Official Review · Reviewer_1YVE · 2025-10-27

**Soundness:** 2
**Presentation:** 2
**Contribution:** 2
**Rating:** 4
**Confidence:** 3

**Summary:**

This paper proposes WaveAttNet, a wavelet-guided attention network for spiking image reconstruction, designed to tackle noise and motion blur. The core ideas are a Wavelet-Guided Attention (WGA) module for frequency-aware denoising and a Multi-scale Temporal Attention (MTA) module to handle motion. While the proposed method achieves strong results on some metrics, it suffers from critical flaws, including extremely high computational cost, uncompetitive performance-efficiency trade-offs compared to recent SOTA models.

**Strengths:**

1. The main contribution, the Wavelet-Guided Attention (WGA) module, is a novel and interesting idea. Using the wavelet transform to separate noise and signal in the frequency domain and then applying attention for filtering is an elegant approach to the denoising problem in spike-based vision.

2. The method achieves state-of-the-art results on several key reconstruction metrics (e.g., PSNR, LPIPS) on the tested datasets, demonstrating the potential of the proposed technical approach.

**Weaknesses:**

1. The paper's most critical flaw is its massive computational cost and the complete lack of relevant efficiency metrics. The reported FLOPs (>1000 G) are extremely high, which strongly suggests that the actual inference latency and energy consumption would be impractical for real-world applications. However, the paper fails to report either of these crucial metrics (e.g., SynOps for energy, or milliseconds per inference for latency), which are standard for evaluating neuromorphic systems.

2. When placed in the context of the latest research, the method's performance-efficiency trade-off is not competitive. The paper lacks a convincing argument for why its high-cost, high-performance approach is superior to some more balanced alternatives.

3. The paper lacks critical details needed to reproduce the results. For instance, the architecture of the "CNN-based extractor" within the MTA module is not specified, nor is the number of parallel branches (N) used in the MTA. Key hyperparameters and the specific configuration of the network backbones are also missing.

**Questions:**

1. How do you justify the extremely high computational cost (FLOPs) of WaveAttNet?

2. Given the extremely high FLOPs, could you please provide the actual energy consumption (e.g., in SynOps or measured energy) and inference latency of your model? How do these crucial metrics compare to other SOTA methods?

3. How do you justify the practical relevance of a method with such a high computational cost in the context of neuromorphic computing, which prioritizes efficiency?

4. Have you considered replacing the heavy CNN backbone with a more efficient architecture, such as a lightweight Transformer, to create a better balance between performance and efficiency?

---

### Official Review · Reviewer_XNPJ · 2025-10-28

**Soundness:** 2
**Presentation:** 3
**Contribution:** 2
**Rating:** 2
**Confidence:** 5

**Summary:**

This paper proposes WaveAttNet, a wavelet-guided attention network for reconstructing images from spike camera data. The method introduces two main components: a Wavelet-Guided Attention (WGA) module for frequency-aware noise suppression, and a Multi-scale Temporal Attention (MTA) module for handling motion blur and low-light artifacts. The authors perform experiments on both synthetic and real-world spike datasets.

**Strengths:**

1. The paper is well-structured and clearly written, making the proposed method and experiments easy to follow.
2. The work provides extensive experimental results on both synthetic and real-world captured datasets.

**Weaknesses:**

1. The paper builds its narrative around "short exposure" and "long exposure" in spike cameras, which is conceptually misleading. Spike cameras operate via continuous photon accumulation without explicit exposure intervals. This may confuse readers and weaken the methodological motivation.
2. The MTA module bears a strong resemblance to existing multi-scale temporal fusion strategies, such as those in Spk2ImgNet. The authors should better clarify the novelty of MTA compared to prior modules.
3. The comparison lacks several recent and relevant methods, such as BSF[1] and STIR[2]. It is also unclear whether the authors used official pre-trained models or retrained all baselines under the same settings—this should be clarified for fair comparison.
[1] Boosting Spike Camera Image Reconstruction from a Perspective of Dealing with Spike Fluctuations. CVPR 2024.
[2] Spatio-Temporal Interactive Learning for Efficient Image Reconstruction of Spiking Cameras. NeurIPS 2025.
4. Although WaveAttNet achieves competitive results, it has the highest FLOPs among the compared methods. The absence of runtime comparisons makes it difficult to assess its practical efficiency. The performance gain may partly stem from increased model capacity rather than architectural superiority.
5. The visual comparisons (Figs. 2 and 3) are not sufficiently convincing. Key regions should be zoomed in for better inspection. In addition, it appears that the exposure parameters differ across the methods.
6. The ablation experiments are somewhat high-level. For instance, Table 5 shows a performance drop without MTA, but it is unclear whether this is due to the removal of the module or the resulting reduction in model capacity. More targeted experiments are needed to validate that WGA and MTA indeed address noise and blur separately.
7. The Real-Captured dataset is described as combining recVidarReal2019 and momVidarReal2021, but it is unclear whether all sequences from these datasets were used or only a curated subset. This could affect the generalization claims.
8. WGSE is a method included in the comparisons, which also leverages the Discrete Wavelet Transform for spike stream processing. Maybe the authors could provide a clear and detailed discussion on how WGA fundamentally differs from or advances beyond the wavelet-based approach in WGSE.

**Questions:**

Please refer to the weaknesses.

---

### Official Review · Reviewer_pewW · 2025-10-30

**Soundness:** 2
**Presentation:** 2
**Contribution:** 2
**Rating:** 4
**Confidence:** 2

**Summary:**

This paper proposes WaveAttNet, a novel CNN-based architecture for reconstructing high-quality images from spike camera data. The core innovation lies in two dedicated modules: a Wavelet-Guided Attention (WGA) module that uses Discrete Wavelet Transform (DWT) and attention to perform frequency-aware noise suppression, and a Multi-scale Temporal Attention (MTA) module that fuses features from multiple temporal scales to mitigate the trade-off between noise (in short exposures) and motion blur (in long exposures).

**Strengths:**

The integration of wavelet transforms for explicit frequency-domain processing is a compelling and technically sound approach for spike data.
The paper includes a thorough empirical evaluation across two distinct datasets (synthetic and real-world) and against a wide array of recent and relevant baseline models.

**Weaknesses:**

The authors openly acknowledge in the conclusion (and Fig. 4) that their model has the highest computational cost (FLOPs) among all compared methods. A deeper discussion of this trade-off earlier in the paper, and perhaps a comparison of inference time (not just FLOPs), would provide a more complete picture.

It would be valuable to see an analysis or visualization of what the MTA's attention maps learn to prioritize. Can you provide any qualitative analysis or visualization of the attention maps learned by the Multi-scale Temporal Attention (MTA) module?

The SNN (SSIR) and Mamba (Spk2ImgMamba) baselines perform surprisingly poorly on the synthetic Spike-REDS dataset. Beyond the explanation of brightness issues, do you have further insight into why these architectures, which are well-suited for temporal data, struggle so significantly with this task?

The paper positions itself as a CNN-based model, which is fine, but the introduction and related work heavily emphasize the advantages of Transformers for long-range dependencies. Given that the proposed modules (especially MTA) effectively capture multi-scale temporal context, a short discussion on how this design compares to or complements a Transformer's self-attention mechanism could be insightful.

**Questions:**

See weakness above.

---

### Official Review · Reviewer_e7Be · 2025-10-31

**Soundness:** 2
**Presentation:** 2
**Contribution:** 1
**Rating:** 2
**Confidence:** 5

**Summary:**

The paper proposes a wavelet attention-based network for spike image reconstruction. A wavelet-guided attention (WGA) module is introduced, and a multi-scale temporal attention (MTA) module is proposed. Experiments are conducted on the Spike-REDS and Real-Captured benchmarks for comparison.

**Strengths:**

1. It is good that the experiment parts attach the parameters and FLOPS comparisons.

**Weaknesses:**

1. The paper lacks novelty. As the task of spike image reconstruction has been explored in many papers, the network proposed in the paper is a kind of toy architecture. Apart from the DWT/IDWT for spikes proposed in the WGSE, the modules introduced in the WGA and MTA are old and commonly-used blocks in deep learning. The paper presents a technical report rather than a research work.
2. I do not consider the MTA as an effective module. The tensors in the "Scale N" already include all information from "Scale 1" to "Scale N-1", and there is no necessity for designing the multi-scale module.
3.  Have all methods for comparison been re-trained with the same setting as the one in the paper?
4. The brightness in Figure 3 is weird. Why results of the proposed method show lower brightness compared to others?
5. The results show no visual advantages compared to others.

**Questions:**

As listed in "Weaknesses".

---

### Note · Authors · 2025-11-12

**Comment:**

We would like to apply for withdrawal. Thank you very much for your time and effort in reviewing our manuscript!

**Withdrawal Confirmation:**

I have read and agree with the venue's withdrawal policy on behalf of myself and my co-authors.